# Zinc Homeostasis: An Emerging Therapeutic Target for Neuroinflammation Related Diseases

**DOI:** 10.3390/biom13030416

**Published:** 2023-02-22

**Authors:** Shunfeng Liu, Nan Wang, Yaqian Long, Zhuan Wu, Shouhong Zhou

**Affiliations:** 1College of Pharmacy, Guilin Medical College, Guilin 541199, China; 2Guangxi Key Laboratory of Brain and Cognitive Neuroscience, Guilin Medical College, Guilin 541199, China; 3Basic Medical College, Guilin Medical College, Guilin 541199, China; 4Department of Radiology, Fifth Clinical Medical College, Guilin Medical College, Guilin 541002, China; 5Central Laboratory, Guangxi Health Commission Key Laboratory of Glucose and Lipid Metabolism Disorders, Second Affiliated Hospital of Guilin Medical College, Guilin 541199, China

**Keywords:** zinc homeostasis, Zn^2+^, neuroinflammation, synaptic zinc, neuroinflammation-related diseases

## Abstract

Zinc is an indispensable trace element in the human body and plays an important role in regulating normal growth and development. Zinc homeostasis in the central nervous system is closely related to the development of neuroinflammation, and synaptic zinc homeostasis disorders affect zinc homeostasis in the brain. Under the condition of synaptic zinc homeostasis, proper zinc supplementation improves the body’s immunity and inhibits neuroinflammation. Synaptic zinc homeostasis disorder in the brain promotes the occurrence and development of neuroinflammation. Cerebral ischemia and hypoxia cause a massive release of synaptic Zn^2+^ into the synaptic cleft, resulting in neurotoxicity and neuroinflammation. Synaptic zinc homeostasis disorder is a high-risk factor for neurodegenerative diseases. Maintaining cerebral zinc homeostasis suppresses the progression of neuroinflammation-mediated neurodegenerative diseases. This article reviews the relationship between brain zinc homeostasis and neuroinflammation and proposes that maintaining synaptic zinc homeostasis prevents neuroinflammation.

## 1. Introduction

Neuroinflammation is an inflammatory response that occurs in the central nervous system (CNS). Because the blood-brain barrier (BBB) prevents a large number of peripheral immune cells from entering the brain, neuroinflammation is different from peripheral inflammation, and microglia are the main functional cells in the CNS involved in the inflammatory response. Resting-state microglia are highly branched and monitor the state of other cells in the brain [1]. Under stimulation, resting microglia become polarized to form an amoeba-like shape and become activated [1]. Proinflammatory stimulation induces the polarization of microglia to the M1 type, which induces microglial expression and release of interleukin (IL)-1β, IL-6, and tumor necrosis factor-α (TNF-α) [2]. Anti-inflammatory stimulation induces microglial polarization to the M2 type, which induces microglia to express and release IL-4, IL-10 and transforms growth factor-β [2].

Zinc is the second most abundant trace element in the human body, and it exists stably in the form of Zn^2+^. Zinc promotes neurogenesis [3], improves immunity [4], acts as a cofactor for enzymes [5], and participates in synaptic transmission [6] and cell apoptosis [7]. Many proteins in the human body, such as the zinc finger protein, contain Zn^2+^; many enzymes in the body need to combine with Zn^2+^ to play a catalytic role [5]. Therefore, maintaining zinc homeostasis is important for the normal functioning of the human body, and zinc homeostasis disorder may be one of the causes of many diseases, such as cerebral ischemia and hypoxia [8], epilepsy [9], depression [10], and neurodegenerative diseases [11]. However, the mechanism by which zinc homeostasis disorders cause these diseases needs to be further researched.

Synaptic zinc homeostasis disorder affects brain zinc homeostasis. When synaptic zinc is in a steady state, appropriate supplement of zinc improves the body’s immunity and inhibits neuroinflammation [12,13]. When synaptic zinc is disordered, abnormal release of synaptic Zn^2+^ leads to severe neuroinflammation. Severe stresses, such as cerebral ischemia and hypoxia [8], and epilepsy [14], lead to a large amount of synaptic Zn^2+^ being released into the synaptic cleft, which mediates neuronal death and severe neuroinflammation. Neurodegenerative diseases are chronic and progressive diseases with widespread neurotransmitter imbalance. Therefore, maintaining synaptic zinc homeostasis plays an important role in delaying the progression of neurodegenerative diseases [11,15,16]. Neurotoxicity and neuroinflammation caused by disorders of synaptic zinc homeostasis promote the progression of neurodegenerative diseases. This article outlines the relationship between brain zinc homeostasis disorder and neuroinflammation and suggests that maintaining synaptic zinc homeostasis prevents neuroinflammation.

## 2. Zinc Homeostasis and Its Regulatory Mechanism

### 2.1. Body Zinc Homeostasis

Zinc homeostasis in the body is regulated by the gastrointestinal tract by absorption of dietary zinc or excretion of zinc by the kidney based on existing levels [17]. More than 90% of the zinc in the body is stored in the skeletal muscle and skeleton, and less than 3% of the total zinc is found in the brain [17]. Zinc in the body exists stably in the form of Zn^2+^, with one part of the Zn^2+^ being in the binding state and the other part being in the free state. Zinc in cells is divided into three parts: 1. Zn^2+^ combined with proteins, including metallothionein (MT); 2. Zn^2+^ in organelles and vesicles; and 3. Free Zn^2+^ in the cytoplasm [15,18]. Zrt/Irt-like protein (ZIP) (encoded by the SLC39A gene family) has 14 types of zinc transporters. ZIP transports extracellular Zn^2+^ as well as Zn^2+^ in organelles and vesicles to the cytoplasm [17]. ZIP plays an important role in maintaining intracellular zinc homeostasis, and the zinc transporter (ZnT) (encoded by the SLC30A gene family) has 10 ZnT members, among which ZnT9 does not possess the function of zinc transport; hence, only nine types of ZnT transport zinc [17]. The outflow of Zn^2+^ from the cytoplasm to the extracellular space is mainly mediated by ZnT1, while other zinc transporters mediate the transport of Zn^2+^ from the cytoplasm to organelles or vesicles [17]. ZnT in the cell membrane is mainly distributed in the organelle membrane, which mediates the uptake of Zn^2+^ by Golgi bodies, lysosomes, and vesicles [17]. MT is a protein rich in cysteine, which scavenges free radicals [19]. MT reduces the free zinc content in cells by combining with Zn^2+^ in the cytoplasm and also releases Zn^2+^ when needed [19]. Therefore, MT is a key protein that regulates cellular zinc homeostasis.

### 2.2. Brain Zinc Homeostasis and Synaptic Zinc

Zinc in the brain is mainly located in glutamate neurons, most of which are highly enriched in Zn^2+^ [6]. Highly enriched Zn^2+^ in the nerve endings of glutamate neurons is released into the synaptic cleft when nerve impulses arrive and enter the postsynaptic membrane, thus regulating the physiological function of the brain [6]. Therefore, an imbalance in synaptic zinc homeostasis significantly affects zinc homeostasis in the brain. Zn^2+^ binds to N-methyl-d-aspartate (NMDA) and γ-aminobutyric acid type A (GABA-A) receptors, thereby affecting excitatory and inhibitory synaptic transmission in the brain [6]. Synaptic transport of Zn^2+^ involves the release of Zn^2+^ from the presynaptic membrane to the postsynaptic membrane and into postsynaptic neurons.

ZnT3 mediates the transfer of Zn^2+^ from cells to synaptic vesicles [20,21], and synaptic vesicles containing Zn^2+^ combine with the presynaptic membrane to release Zn^2+^ into the synaptic cleft. The pathways of Zn^2+^ entering the postsynaptic neurons in the synaptic cleft include the following [6]: 1. ZIP-mediated transport; 2. Voltage-gated calcium channel (VGCC)-mediated transport; 3. Ca^2+^-and Zn^2+^-permeable GluR2-lacking AMPA (α-amino-3-hydroxy-5-methyl-4-isoxazole propionic acid) receptors (AMPAR_Ca-Zn_)-mediated transport; 4. NMDA receptor-mediated transport; and 5. Na^+^/Zn^2+^ exchanger-mediated transport [22]. The main transport protein of Zn^2+^ efflux is ZnT1, whereas MT stores Zn^2+^ and dynamically regulates zinc homeostasis in cells. The release of zinc from presynaptic to postsynaptic neurons increases the level of oxidative stress in postsynaptic neurons, leading to neuronal death [16,23] (as shown in Figure 1).

## 3. Matrix Metalloproteinases and Neuroinflammation

The catalytically active region of matrix metalloproteinase (MMP) contains a zinc-binding site, which is a zinc-dependent endopeptidase. Under the regulation of inflammatory cytokines and reactive oxygen species (ROS), neurons, astrocytes, and microglia express and release MMP [24,25]. MMP’s function has two sides. MMP promotes tissue remodeling and repair. However, excessive activation also leads to tissue damage. Furthermore, overactivation of MMP destroys the BBB, leading to peripheral immune cells entering the brain and participating in the inflammatory response in the CNS. There is an interaction between neuroinflammation and MMP. IL-1β induces pericyte expression and secretion of MMP-9 by activating the nuclear factor-κB (NF-κB) signaling pathway, and high expression of MMP-9 damages the integrity of the BBB [26]. An in vitro study in which neurons and astrocytes were isolated from mice with Alzheimer’s disease (AD) showed that knockout of the MMP-24 gene significantly reduced IL-1β and β-amyloid (Aβ) in cells [27]. Lipopolysaccharide (LPS) activates the NF-κB signaling pathway by increasing ROS content in astrocytes, thereby upregulating MMP-9 expression [28].

Zinc promotes MMP-mediated neuroinflammation. MMP activation requires Zn^2+^ binding to the catalytically active region of MMP. Treatment with the Zn^2+^ chelator clioquinol (CQ) significantly inhibited MMP-2 and MMP-9 activity in hippocampal neurons, indicating that Zn^2+^ is critical for MMP activation [29]. Treatment of mice with minocycline, an inhibitor of microglial activation, inhibits rotenone-induced MMP-2 and MMP-9 activation, indicating that microglial activation is a key factor mediating MMP activation [30]. Zinc chloride (ZnCl_2_) treatment enhanced the effect of LPS-induced microglia to release proinflammatory factors, but ROS scavenger inhibited the effect of ZnCl_2_, suggesting that Zn^2+^ promoted microglial expression and release of proinflammatory factors by mediating ROS generation [31]. Zn^2+^ increased the intracellular ROS content by activating reduced nicotinamide adenine dinucleotide phosphate oxidase (NOX) in microglia [8,32]. Zn^2+^ treatment significantly increased the content of ROS in neurons [33] and enhanced the effect of LPS-induced ROS production in astrocytes [34]. Persistently high concentrations of intracellular and extracellular Zn^2+^ would promote increased production of ROS and proinflammatory factors, especially in microglia, which mediate immune responses [32,35,36]. The generated ROS and proinflammatory factors increase the activity and expression of MMP, which further promotes neuroinflammation-mediated CNS injury (Figure 1).

## 4. Zinc Homeostasis and Neuroinflammation

### 4.1. Zinc Homeostasis Has a Role in Regulating Neuroinflammatory Responses

Zinc sulfate (ZnSO_4_) downregulated the expression of inducible NO synthase (iNOS), TNF-α, and IL-6 induced by LPS (100 ng/mL) and significantly reduced the content of ROS in cells [37]. A20 is a negative regulator of NF-κB signaling and contains a zinc-finger domain. ZnSO_4_ inhibits NF-κB signaling by upregulating A20 expression in BV2 cells, thereby inhibiting LPS-induced neuroinflammation [37]. Arginase 1 (Arg-1) is a marker of M2 polarization in microglia. IL-4 induces microglia to express Arg-1 mRNA and increases the cellular zinc in cells [38]. The intracellular Zn^2+^ chelator TPEN upregulated the expression of Arg-1 mRNA in microglia induced by IL-4, indicating that intracellular free Zn^2+^ negatively regulates the expression of Arg-1 [38]. In summary, intracellular Zn^2+^, in cases of zinc homeostasis, inhibits microglial polarization to the M2 type. Under conditions of zinc homeostasis, Zn^2+^ not only inhibited the expression of proinflammatory factors in microglia but also inhibited the expression of anti-inflammatory factors, playing a role in regulating the neuroinflammatory response.

Zinc supplementation inhibited the activation of microglia in the cerebral cortex of rats fed a high sugar and fat diet, and downregulated the toll-like receptor 4 expression in the rat hippocampus. A high sugar and fat diet damaged the cognitive function of rats, and zinc supplementation significantly improved their cognitive function [4]. Therefore, zinc supplementation improved cognitive function in rats by reducing neuroinflammation. A high-fat diet induced high expression of IL-1β in the cerebral cortex and hippocampus of rats, while zinc supplementation significantly reduced the expression of IL-1β and enhanced novel object recognition in the rats [4]. Thus, zinc supplementation inhibited high-fat diet induced neuroinflammation. Maternal zinc supplementation downregulated the expression of the proinflammatory factors IL-6, IL-1β, and TNF-α in the prefrontal cortex (PFC) of LPS-exposed adult male offspring and decreased the expression of NF-κB in the PFC [13].

### 4.2. Zinc Homeostasis Imbalance

#### 4.2.1. Low Zinc Promotes Neuroinflammation

TPEN significantly upregulated IL-1β expression in macrophages treated with LPS (1 μg/mL), whereas ZnCl_2_ treatment reversed the effects of TPEN [39]. TPEN enhanced the release of IL-1β from macrophages induced by LPS [40]. Compared to monocytes cultured at normal Zn^2+^ concentrations, zinc-deficient monocytes treated with LPS showed significantly upregulated TNF-α and IL-1β expressions, indicating that zinc-deficient monocytes were more sensitive to LPS-induced inflammation [41]. A low-zinc diet or inhibition of zinc absorption changed the intestinal microbiota composition and intestinal function of pregnant mice, upregulating the expression of IL-6 in the brain and activating astrocytes in the hippocampus [42]. Therefore, zinc deficiency may contribute to neuroinflammation by altering intestinal function and intestinal microbiota composition. Prenatal zinc-deficient mice exhibit autism spectrum disorder. A study found that the brains of prenatal zinc-deficient mice showed a neuroinflammatory response, suggesting that prenatal zinc deficiency-mediated neuroinflammation may be the cause of autism in mice [43].

#### 4.2.2. Zinc Accumulation Accelerates Neuroinflammation

Zn^2+^ increased the content of ROS in microglia by activating NOX in microglia; then, ROS activated the extracellular signal-regulated kinase (ERK) signaling pathway to upregulate the expression of transient receptor potential melastatin 2 (TRPM2) on the cell membrane, which mediates Ca^2+^ influx [32]. Therefore, microglia were overloaded with Ca^2+^, leading to cell death. Zeolite-based nanomaterials (Zbn) adsorbed excessive Zn^2+^ and cleared ROS in the brain of rats after cerebral ischemia, inhibiting the activation of microglia [44]. Therefore, Zbn provided neuroprotection by reducing the content of Zn^2+^ in the brain. TPEN chelation of Zn^2+^ inhibited the activation of microglia induced by ZnCl_2_, indicating that Zn^2+^ activated microglia and promoted neuroinflammation [35]. Hypoosmotic stress increased the level of extracellular free Zn^2+^ by upregulating the expression of ZnT1 in astrocytes. Microglia were activated by culture medium obtained by treating astrocytes with hypoosmotic stress, whereas extracellular Zn^2+^ chelator CaEDTA inhibited microglial activation [45]. Therefore, hypoosmotic stress promoted the release of Zn^2+^ from astrocytes by upregulating the expression of ZnT1 in astrocytes, thus activating microglia. Zn^2+^ promoted LPS-induced production of ROS in astrocytes. Zn^2+^ significantly increased NO production in astrocytes induced by LPS (1 μg/mL) through the p38-mitogen-activated protein kinase signaling pathway [34]. Zn^2+^ treatment reduced the levels of glutathione (GSH), superoxide dismutase (SOD), and catalase (CAT) in neurons; thus, leading to an increased level of oxidative stress [46]. In conclusion, Zn^2+^ promoted oxidative stress and neuroinflammation (Figure 1).

### 4.3. Severe Inflammatory Stress Leads to Zinc Homeostasis Imbalance

Intraperitoneal injection of N-acetyl-L-cysteine (NAC) in rats significantly reduced neuronal death and zinc accumulation induced by cerebral ischemia. NAC downregulated the expression of the non-selective cationic channel TRPM2 in neurons, thereby inhibiting the activation of astrocytes and microglia [47]. Therefore, NAC may reduce the entry of Zn^2+^ into neurons by downregulating the expression of TRPM2, thus reducing the neuroinflammation mediated by Zn^2+^. Another study showed that treatment with interferon γ (INFγ) increased Zn^2+^ content in neurons and upregulated IL-1β, IL-6, and TNF-α expressions in microglia, whereas TRPM2 channel inhibitors reversed the effects of INF γ, indicating that INFγ increased intracellular Zn^2+^ content by activating TRPM2 channels and promoted neuroinflammation [48]. Therefore, severe inflammatory stress led to an excessive influx of Zn^2+^ into cells by upregulating the expression of TRPM2, leading to neuroinflammation.

## 5. Zinc Homeostasis and Neuroinflammatory Diseases

### 5.1. Zinc Supplementation Promotes the Recovery of Spinal Cord Injury (SCI)

Intraperitoneal injection of zinc gluconate (ZnG) in mice upregulated the expression of Bcl-2 and downregulated the expression of Bax in the injured spinal cord, indicating that ZnG inhibited cell apoptosis in the injured spinal cord [49]. TPEN chelating Zn^2+^ promoted apoptosis of hippocampal neurons, ERK inhibitor significantly increased TPEN-induced apoptosis of neurons, and ERK agonist reversed the effect of TPEN [50]. Therefore, Zn^2+^ reduced neuronal apoptosis by activating the ERK signaling pathway. ZnG promoted macrophages to express and release granulocyte colony stimulating factor (G-CSF) [7]. Intraperitoneal injection of ZnG promoted the production of G-CSF in the spinal cord tissue of spinal cord injury (SCI) mice and inhibited the apoptosis of nerve cells, but G-CSF neutralizing antibody reversed this effect [7]. Therefore, ZnG inhibited neuronal apoptosis in the injured spinal cord by promoting G-CSF expression. Treatment with methylprednisolone (MP) reduced neuronal death in rats with SCI, downregulated iNOS, IL-6, and IL-10 expressions in the spinal cord, and inhibited microglial activation. In vitro experiments showed that MP treatment significantly increased Zn^2+^ and ZIP8 levels in microglia and reversed LPS-induced NF-κB expression in microglia [51]. Therefore, MP may inhibit the NF-κB pathway by promoting the accumulation of Zn^2+^ in microglia, reducing the release of inflammatory factors and subsequently reducing SCI.

Intraperitoneal injection of ZnG into SCI mice downregulated the expression of NOD-like receptor protein 3 (NLRP3), apoptosis-associated speck-like protein containing a CARD (ASC), caspase-1, and IL-1β in the spinal cord, indicating that zinc treatment inhibited the NLRP3 inflammatory pathway. An in vitro study showed that ZnG inhibited the activation of the LPS-induced NLRP3 inflammatory pathway in microglia [49]. Therefore, Zn^2+^ reduced the expression and release of inflammatory factors by inhibiting the NLRP3 inflammatory pathway in microglia and accelerating the repair of spinal cord tissues. Intraperitoneal injection of ZnG into SCI mice increased the content of Beclin-1 and the ratio of microtubule-associated protein 1 light chain 3-II/I and reduced the expression of P62. Therefore, ZnG promoted autophagy in nerve cells in the injured spinal cord [49]. The expressions of IL-1β, IL-6, and the NLRP3 inflammasome upregulated in the injured spinal cord, and intraperitoneal injection of ZnG promoted autophagy and inhibited the expression of the NLRP3 inflammasome in SCI mice [52]. An in vitro study showed that ZnG promoted autophagy of microglia and inhibited the expression of the NLRP3 inflammasome, and an autophagy inhibitor reversed the effect of zinc on inhibiting the expression of the NLRP3 inflammasome in microglia. A ubiquitin inhibitor reversed the effect of zinc in promoting NLRP3 inflammasome degradation in microglia, suggesting that zinc promotes NLRP3 degradation through ubiquitination modification [52]. In conclusion, zinc reduced NLRP3 inflammasome content through autophagy and ubiquitination, subsequently promoting repair in SCI mice (Figure 2).

### 5.2. Zinc Supplement Has Antidepressant Effect

Clinical research has shown that the decrease in zinc concentration in peripheral blood is related to the occurrence and development of depression [53], and zinc supplementation enhances the therapeutic effect of antidepressants [10,54,55]. A low-zinc diet significantly increased the immobility time of mice in the forced swim test [56] and the tail suspension test [57], and decreased the preference for sugar and water [58]. Therefore, the reduction of zinc intake in the body could lead to depression. The combination of ZnCl_2_ and the natural peptide Cyclo-(His Pro) generated zinc plus Cyclo-(His-Pro) (ZC). Feeding rats with ZC enhanced the body’s absorption of zinc and significantly increased the level of vesicular zinc and neurogenesis in the hippocampus [59]. ZnCl_2_ treatment upregulated the expression of nestin in induced pluripotent stem cells (iPSCs), whereas TPEN treatment inhibited the expression of nestin in iPSCs, indicating that Zn^2+^ promoted the differentiation of iPSCs. ZnCl_2_ treatment promoted ERK and signal transducer and activator of transcription (STAT) phosphorylation, but ERK inhibitors inhibited STAT3 phosphorylation and downregulated nestin expression, indicating that Zn^2+^ promoted iPSCs differentiation through the ERK-STAT3 signal pathway [60]. Patients with depression show decreased hippocampal neurogenesis, and zinc supplementation promotes hippocampal neurogenesis and has anti-depressive effects. Therefore, zinc supplementation may play an antidepressant role by enhancing neurogenesis.

### 5.3. Zinc Accumulation Promotes the Development of Acute Cerebral Ischemia

After global cerebral ischemia, the release of Zn^2+^ from the presynaptic membrane to the postsynaptic membrane led to increased Zn^2+^ content in postsynaptic neurons, followed by neuronal degeneration, while Zn^2+^ chelation reduced neuronal degeneration, indicating that the massive release of Zn^2+^ from the presynaptic membrane is neurotoxic after global cerebral ischemia [61]. Hypobaric hypoxia led to a protein imbalance related to zinc homeostasis, upregulated the expression of TNF-α and iNOS, and also led to neuronal apoptosis in the hippocampus. Zn^2+^ chelators significantly reversed these effects, indicating that zinc promoted neuroinflammation and apoptosis during hypobaric hypoxia [62]. Chelating Zn^2+^ by intravenous etidronate-zinc complex (Eti-Zn) inhibited the activation of microglia and decreased the expressions of IL-1β, IL-6, and TNF-α after cerebral ischemia and hypoxia. Eti–Zn also activated the NF-κB signaling pathway, suggesting that Eti–Zn reduced microglial activation by inhibiting the NF-κB signaling pathway, thereby inhibiting neuroinflammation [63]. ZnCl_2_ significantly induced ROS production in microglia and also enhanced the effects of LPS on the expression of the proinflammatory factors IL-1β, IL-6, and TNF-α [8]. Cerebral ischemia induced microglia to transform into the M1 type and upregulated proinflammatory factor expression in the mouse hippocampus, whereas CaEDTA inhibited its expression in mice after cerebral ischemia, suggesting that chelating Zn^2+^ attenuated post-ischemic brain injury by inhibiting neuroinflammation [8]. Pretreatment with a high concentration of zinc enhanced the effect of LPS-induced proinflammatory factor secretion in microglia, but this effect was reversed by TPEN and ROS scavengers [31]. Preinjection of CaEDTA into the lateral ventricle inhibited M1 type polarization of microglia in the hippocampus, downregulated the expression of IL-1β, IL-6, and TNF-α, and improved cognitive function in mice after ischemia and reperfusion [31]. In conclusion, zinc accumulation promoted transformation of microglia into a proinflammatory M1 type via ROS, thereby impairing cognitive function in mice after acute cerebral ischemia (Figure 2).

### 5.4. Zinc Homeostasis Disorder Promotes the Development of Epilepsy

Synaptic zinc homeostasis disorder increased susceptibility to epilepsy [9,64]. Synaptic zinc homeostasis disorder led to an imbalance in central excitation and inhibition, which causes abnormal nerve conduction and triggers epilepsy [9,65]. Knockout of MT3 in mice significantly reduced kainate-induced zinc accumulation, neuronal death, and epileptic seizures in the hippocampal CA1 region [14]. Intracerebroventricular injection of CaEDTA reduced kainate-induced hippocampal neuronal death and epileptic seizures in wild-type mice, indicating that Zn^2+^ overload is a vital cause of neuronal death and epilepsy [14]. Knockout of Zip1 and Zip3 genes inhibited zinc absorption by pyramidal neurons in the hippocampus and kainic acid-induced epilepsy in mice, thus reducing neuronal damage [66]. Therefore, chelating Zn^2+^ may be a measure to prevent epileptic seizures. Transient receptor potential cation channel 5 (TRPC5) was involved in abnormal epileptic discharges and mediated neuronal death. Treatment of cortical neurons with hydrogen peroxide led to zinc-dependent TRPC5 opening, which triggered Ca^2+^ influx and caused neuronal death and seizures [67]. Therefore, zinc plays an important role in the process of seizure-mediated neuronal death. Ganaxolone (GX) inhibited seizures by acting on extrasynaptic GABA-A receptors. Zn^2+^ selectively blocked extrasynaptic GABA-A receptors in the hippocampus, thereby attenuating the antiepileptic effect of GX [68] (Figure 2).

### 5.5. Zinc Homeostasis Imbalance Promotes Alzheimer’s Disease Progression

Daily intraperitoneal injections of the Zn^2+^ chelator CQ for 2 weeks significantly decreased cognitive function in young mice [29]. Knockdown of ZnT3 significantly impaired cognitive function in mice [69]. Knockout of ZnT3 significantly reduced zinc levels in the mouse hippocampus, led to abnormal pre- and post-synaptic protein expressions, and affected learning and memory. CQ treatment reversed these phenomena in ZnT3 knockout mice, indicating that zinc homeostasis disorder led to cognitive impairment [70]. Intracellular and extracellular ROS increased the content of free Zn^2+^ [71]. ROS increased the content of Zn^2+^ in the neurons of AD mice. Intraperitoneal injection of pyruvate reduced the expression of ROS and the content of Zn^2+^ in neurons of AD mice and improved their cognitive function [72]. Therefore, pyruvate reduced the content of Zn^2+^ in the brain by downregulating the level of ROS in neurons, thereby improving the cognitive function of AD mice.

ZnAβ generated by the combination of Zn^2+^ and Aβ significantly inhibited long-term potentiation in the mouse hippocampus and activated microglia, and Zn^2+^ also enhanced the toxicity of Aβ to neuroblastoma cells [73]. Thus, ZnAβ is more neurotoxic than Aβ. Tau mice are transgenic mice containing human tau protein gene. High concentrations of zinc significantly increased the phosphorylation level of tau in tau mice, the number of tau tangles in the hippocampus, and memory impairment in tau mice [74]. Intracerebroventricular injection of ZnSO_4_ caused hyperphosphorylation of tau in the hippocampus, promoted oxidative stress injury, and impaired learning and memory function in rats. The nuclear factor E2 related factor 2 (Nrf2)/ heme oxygenase-1 (HO-1) pathway was inhibited by intracerebroventricular injection of ZnSO_4_ in rats and ZnSO_4_ treatment of SH-SY5Y cells [33]. In conclusion, ZnSO_4_ promoted tau phosphorylation by inhibiting the Nrf2/HO-1 pathway, thereby damaging the cognitive function of rats. Zn^2+^ enhanced tau neurotoxicity by inducing tau hyperphosphorylation and conformational changes. CQ reversed the effect of Zn^2+^ on tau, indicating that Zn^2+^ is the key to tau neurotoxicity [75]. Zn^2+^ enhanced the toxic effects of Aβ and promoted tau phosphorylation, indicating that restoring zinc homeostasis in the brain can prevent and treat AD [76] (Figure 2).

### 5.6. Zinc Led to Parkinson’s Disease Mediated by Inflammatory Response and Oxidative Stress

Intraperitoneal injection of high concentrations of ZnSO_4_ increased the content of Zn^2+^ in the substantia nigra and striatum, promoted oxidative stress, reduced the level of dopamine in the striatum, and reduced the spontaneous locomotor activity of rats [77]. Intraperitoneal injection of high concentrations of ZnSO_4_ in rats caused α-synuclein (α-syn) aggregation in the substantia nigra dopaminergic neurons [78]. Kufor-Rakeb syndrome (KRS) is a genetic disease caused by ATP13A2 (PARK9) mutations that manifests as Parkinson’s disease (PD). Knockdown of the PARK9 gene in primary cortical neurons caused the accumulation of α-syn, and Zn^2+^ treatment further increased α-syn accumulation, indicating that Zn^2+^ promoted the progression of PD [79]. Intraperitoneal injection of high concentrations of ZnSO_4_ in rats reduced the number of dopamine neurons and dopamine content, and increased the activity of NOX in the substantia nigra and striatum. The inhibition of microglial activation by minocycline reversed the effects of ZnSO_4_, suggesting that a high concentration of Zn^2+^ caused dopamine neuron damage by activating microglia to release inflammatory factors; thus, participating in the occurrence of PD [80]. The intraperitoneal injection of high concentrations of ZnSO_4_ in adult rats reduced the number of substantia nigra dopamine neurons and the content of striatal dopamine, increased the expression of oxidative stress and inflammatory factors, and caused PD symptoms. ZnSO_4_ treatment in newborn and adult rats further aggravated the above effects, indicating that a high concentration of Zn^2+^ promoted the occurrence and development of PD [81]. It was also found that high concentrations of Zn^2+^ promoted the release of proinflammatory factors from microglia in the substantia nigra by activating the NF-κB pathway and also promoted apoptosis [81] (Figure 2).

AMPA injection into the substantia nigra significantly increased the amount of free Zn^2+^ and the loss of dopamine neurons in the substantia nigra and caused movement disorder in rats. Zn^2+^ chelator reversed these effects, suggesting that AMPAR_Ca-Zn_ mediated Zn^2+^ influx and damaged nigral dopamine neurons, leading to PD [82]. 6-OHDA impaired dopamine neurons in the substantia nigra and striatum by promoting extracellular Zn^2+^ to flow into dopamine neurons [83], while blocking Zn^2+^ influx into dopamine neurons inhibited paraquat-induced dopamine neuron loss [84]. In conclusion, a large influx of extracellular Zn^2+^ into dopaminergic neurons promoted the development of PD. Synaptic Zn^2+^ release led to a rapid increase in extracellular Zn^2+^ concentration, so maintaining synaptic zinc homeostasis is a potential strategy to combat PD [15,82].

### 5.7. Zinc Accumulation Promotes the Occurrence and Development of Multiple Sclerosis

ZnT3 on the synaptic vesicle membrane transports zinc into the vesicles for storage [20,21]. Knockdown of ZnT3 reduced Zn^2+^ release into the synaptic cleft, significantly inhibited spinal white matter demyelination, reduced the inflammatory response in mice with experimental autoimmune encephalomyelitis (EAE), and significantly inhibited the activation of MMP-9 and the formation of zinc plaques [3]. 1H10 is an inhibitor of AMP-activated protein kinase (AMPK) phosphorylation, and it was experimentally demonstrated that 1H10 also has the ability to chelate Zn^2+^. Intraperitoneal injection of 1H10 in mice inhibited EAE-induced demyelination and microglia/macrophage activation, inhibited EAE-induced MMP-9 activation and zinc plaque formation and alleviated multiple sclerosis (MS) [85]. Therefore, inhibition of AMPK phosphorylation or chelation of Zn^2+^ may be an effective target for MS treatment. Zn^2+^ damaged myelin sheath by activating MMP-9 to destroy the BBB and activating microglia to release proinflammatory cytokines [16] (Figure 2).

## 6. Summary and Outlook

Synaptic zinc homeostasis plays an important role in maintaining brain zinc homeostasis and CNS stability. When synaptic zinc is in a homeostatic state, preventive zinc supplementation can improve the body’s immunity and reduce the production of inflammatory factors in the brain and is conducive to the utilization of zinc in the body [12,13]. Under intense stress conditions, such as cerebral ischemia [8], epilepsy [14], and hypoglycemia [86], the presynaptic membrane Zn^2+^ is released at a great rate into the synaptic cleft, resulting in Zn^2+^ overload in the postsynaptic neurons and neurotoxicity. Excessive Zn^2+^ levels in the synaptic cleft also activate extracellular MMP and microglia, promoting neuroinflammation and causing tissue damage [16]. Thus, the use of extracellular Zn^2+^ chelators, reduces neuronal death caused by ischemic hypoxia and epilepsy, and resists neuroinflammation.

Neurodegenerative diseases are closely related to synaptic transmission, such as the imbalance of acetylcholine and dopamine levels in AD and PD patients. Zn^2+^ uptake in vitro and a large amount of Zn^2+^ released by impaired synaptic function can activate microglia and also enhance inflammatory protein (α-syn, tau, Aβ, and MMP) expression and neurotoxicity, thereby driving neuroinflammation-mediated neuronal damage [16,33,73,78]. Therefore, maintaining synaptic zinc homeostasis plays an important role in treating neurodegenerative diseases [11,15,16]. Synaptic zinc homeostasis disorders cause neurotoxicity and further affect nerve transmission, thereby promoting the progression of neurodegenerative diseases. Regulating zinc homeostasis in the early onset of neurodegenerative diseases may delay the disease progression. When neurodegenerative disease progresses to a moderate or severe degree, nerve transmission may become impaired, and zinc intake should be strictly controlled.

Preparations for zinc supplementation have evolved from inorganic zinc agents to organic zinc agents [87]. Inorganic zinc agents have the disadvantages of strong intestinal irritation and low bioavailability [88]. Now, an organic zinc agent is mainly used to improve zinc content in the body, including zinc gluconate, zinc citrate, zinc-enriched yeast and protein-chelated zinc [88,89,90]. The clinical application of zinc chelating agent is chloroiodohydroxyquine, which is mainly used in the treatment of dermatological diseases [91,92]. Some studies showed the neuroprotective effect of chelating Zn^2+^ after cerebral ischemia and hypoxia [8,31,63], and the development of a clinical drug to Zn^2+^ in the brain may have a high application prospect.

Neuroinflammation is a common pathophysiological basis of neurodegenerative diseases and is a major research topic in the field of neurodegenerative diseases [93]. Regulation of neuroinflammation is an important way to prevent and treat neurodegenerative diseases. Zinc homeostasis disorders affect the occurrence and development of neuroinflammation. Therefore, restoring zinc homeostasis by targeting specific proteins that regulate zinc homeostasis may be vital for the prevention and treatment of neuroinflammation-related neurodegenerative diseases. Hence, the role of zinc homeostasis disorders in the development and progression of neuroinflammation has received increasing attention. Further clarification of the molecular mechanism of neuroinflammation caused by zinc homeostasis disorders has profound scientific and clinical significance for the prevention and treatment of neuroinflammation-related neurodegenerative diseases.

## Figures and Tables

**Figure 1 biomolecules-13-00416-f001:**
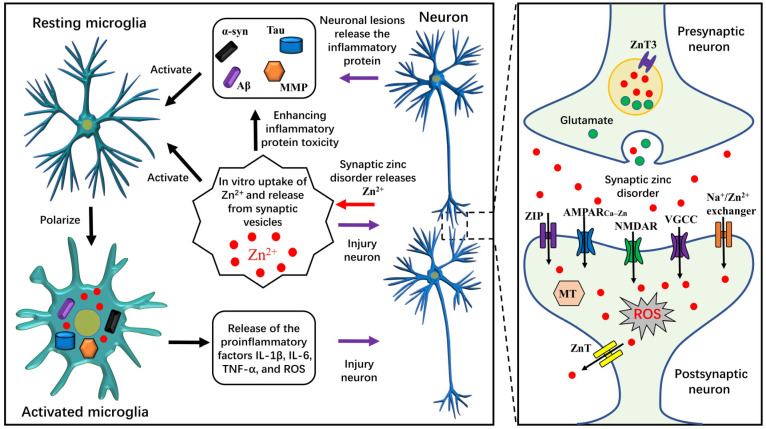
Zinc homeostasis disorder in the brain promotes neuroinflammation. When diseases, such as AD and PD, and conditions, such as cerebral ischemia and hypoxia, lead to synaptic function disorder, synaptic zinc levels are in a state of fragile balance or imbalance. In this case, zinc supplementation increases the extracellular Zn^2+^ and synaptic Zn^2+^ content in the brain, leading to the uncontrolled release of a large amount of Zn^2+^ from the synapse to the extracellular space. Large amounts of extracellular Zn^2+^ damage neurons in several ways: 1. Zn^2+^ flows directly into neurons, thus promoting oxidative stress and causing neuronal death; 2. Zn^2+^ damages neurons by activating microglia to produce proinflammatory factors and ROS; 3. Zn^2+^ damages neurons by enhancing the expression and neurotoxicity of inflammatory proteins (α-syn, Tau, Aβ, and MMP). Therefore, patients with disorders involving abnormal synaptic function should strictly control zinc uptake. α-synuclein (α-syn); Alzheimer’s disease (AD); Parkinson’s disease (PD); β-amyloid (Aβ); matrix metalloproteinase (MMP); interleukin (IL); tumor necrosis factor-α (TNF-α); reactive oxygen species (ROS); zinc transporter (ZnT); Zrt/Irt-like protein (ZIP); Ca^2+^-and Zn^2+^-permeable GluR2-lacking AMPA (α-amino-3-hydroxy-5-methyl-4-isoxazole propionic acid) receptors (AMPAR_Ca-Zn_); N-methyl-d-aspartate receptor (NMDAR); voltage-gated calcium channel (VGCC); metallothionein (MT).

**Figure 2 biomolecules-13-00416-f002:**
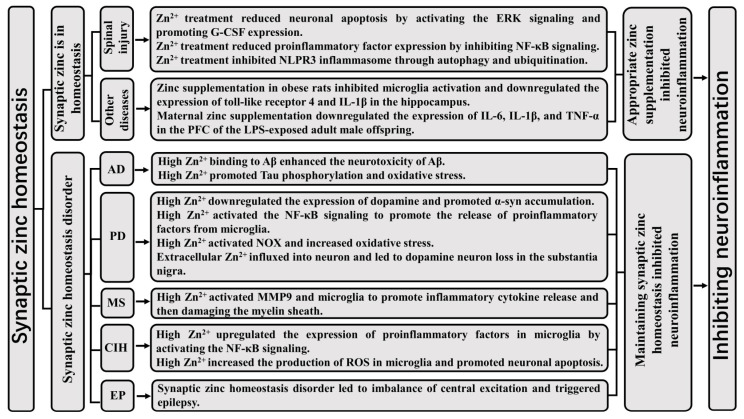
Relationship between synaptic zinc and neuroinflammation. Zinc is an indispensable trace element for maintaining life-preserving activities of organisms. When synaptic zinc is in homeostasis, the appropriate zinc supplementation enhances the body’s utilization of zinc, thereby enhancing the body’s ability to resist neuroinflammation. However, synaptic zinc disorder further damages the synaptic function when suffering from synapse damage-related diseases. Zn^2+^ uptake in vitro and the large amount of Zn^2+^ released by impaired synapses promote neuroinflammation, so maintaining synaptic zinc homeostasis inhibits neuroinflammation. ERK (extracellular signal-regulated kinase); granulocyte colony stimulating factor (G-CSF); nuclear factor-κB (NF-κB); NOD-like receptor protein 3 (NLRP3); interleukin (IL); tumor necrosis factor-α (TNF-α); prefrontal cortex (PFC); lipopolysaccharide (LPS); Alzheimer’s disease (AD); Parkinson’s disease (PD); multiple sclerosis (MS); cerebral ischemia and hypoxia (CIH); epilepsy (EP); β-amyloid (Aβ); α-synuclein (α-syn); reduced nicotinamide adenine dinucleotide phosphate oxidase (NOX); matrix metalloproteinase (MMP); reactive oxygen species (ROS).

## Data Availability

Not applicable.

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
