# Peer review of "Zinc Homeostasis: An Emerging Therapeutic Target for Neuroinflammation Related Diseases"

_biomolecules, 2023, doi:10.3390/biom13030416_

Round 1
Reviewer 1 Report
This review demonstrates that restoration of zinc homeostasis may be emerging therapeutic target for neuroinflammation-related diseases. Recently, a lot of knowledge has been reported about the importance of zinc and what kind of neuroinflammation-related diseases is modified by zinc imbalances, and the way in which it can be used as a therapeutic target is of interest to many readers. However, much of this review merely lists the biological effects of zinc-related substances, and the title does not match the content; it needs to be revised to focus on restoring zinc homeostasis as a novel therapeutic target for neuroinflammation-related diseases
1. The ZO-NPs in Chapter 4.2.2 and ZnG in Chapter 5.1 are not connected to the story that "restoration of zinc homeostasis could be a novel therapeutic target for neuroinflammation-related diseases," but simply show the biological effects of zinc compounds. If it were a finding that administration of zinc as these compounds improved neuroinflammation-related diseases in some way, it would make sense to introduce it in this review, but it merely introduces the toxicity of these compounds. You need to remove topics that are not relevant to the title.
2. I think the part of chapter 3 that presents findings related to Matrix metalloprotease is good, but I think the explanation is insufficient. Regarding the last 5 lines, why does excess zinc increase ROS? Is this and the increase in proinflammatory factor related, independent, or unknown phenomena? I look forward to your research into these areas.
3. Are the zinc preparations (ex. Polaprezinc, Nobelzin (zinc acetate dihydrate) etc) that are currently available likely to work or not? It would be good to have a chapter summarizing such matters. Could we use zinc chelator CQ? If not, what areas need to be improved?
4. After mentioned that zinc plays two roles, I don't think those two are explicitly stated, lines 51 and 446. What is the role of zinc in the body?
Author Response
- The ZO-NPs in Chapter 4.2.2 and ZnG in Chapter 5.1 are not connected to the story that "restoration of zinc homeostasis could be a novel therapeutic target for neuroinflammation-related diseases," but simply show the biological effects of zinc compounds. If it were a finding that administration of zinc as these compounds improved neuroinflammation-related diseases in some way, it would make sense to introduce it in this review, but it merely introduces the toxicity of these compounds. You need to remove topics that are not relevant to the title.
Rep: Thank you for your careful reading and suggestions. We have removed the references concerning the zinc oxide nanoparticles (ZO-NPs). Zinc gluconate (ZnG) is an organic zinc agent and dissociates into Zn2+ in the body. Spinal cord injury is related to neuroinflammation. We first illustrate that Zn2+ inhibit the apoptosis of nerve cells, and then illustrate that zinc ions inhibit the neuroinflammation after spinal cord injury. Therefore, we do not remove the ZnG-related references and content.
- I think the part of chapter 3 that presents findings related to Matrix metalloprotease is good, but I think the explanation is insufficient. Regarding the last 5 lines, why does excess zinc increase ROS? Is this and the increase in proinflammatory factor related, independent, or unknown phenomena? I look forward to your research into these areas.
Rep: We have added some references stating that Zn2+ increases the content of ROS in nerve cell. Microglia are the main functional cells involved in the production and release of proinflammatory factors in the brain. Therefore, Zn2+ will not only increase the content of ROS in microglia, but also promote the expression and release of proinflammatory factors in microglia. ZnCl2 treatment enhanced the effect of LPS-induced microglia to release proinflammatory factors, but ROS scavenger inhibited the effect of ZnCl2, suggesting that Zn2+ promoted microglial expression and release of proinflammatory factors by mediating ROS generation [1]. Zn2+ increased the intracellular ROS content by activating the NADPH oxidase in microglia [2] [3]. The increase of ROS in microglia is related to the expression of proinflammatory factor.
Zn2+ treatment significantly increased the content of ROS in neurons [4] and significantly enhanced the effect of LPS-induced ROS production in astrocytes [5]. Zn2+ leads to an increase in ROS production in neurons and astrocytes, but since they hardly express pro-inflammatory factors, ROS and pro-inflammatory factors are expressed independently in these cells.
- Are the zinc preparations (ex. Polaprezinc, Nobelzin (zinc acetate dihydrate) etc) that are currently available likely to work or not? It would be good to have a chapter summarizing such matters. Could we use zinc chelator CQ? If not, what areas need to be improved?
Rep: Thank you for your advice. We have increased some references in summary and outlook. At present, few zinc chelating agent are used in clinical practice. Preparations for zinc supplementation have evolved from inorganic zinc agents to organic zinc agents [6]. Inorganic zinc agents have the disadvantages of strong intestinal irritation and low bioavailability [7]. Now organic zinc agent are mainly used to improve the zinc content in the body, including zinc gluconate, zinc citrate, zinc-enriched yeast and protein-chelated zinc [7-9]. The clinical application of zinc chelating agent is chloroiodohydroxyquine, which is mainly used in the treatment of dermatological diseases [10, 11]. Some studies showed the neuroprotective effect of chelating Zn2+ after cerebral ischemia and hypoxia [1, 3, 12], and the development of a clinical drug to Zn2+ in the brain may have a high application prospect.
- After mentioned that zinc plays two roles, I don't think those two are explicitly stated, lines 51 and 446. What is the role of zinc in the body?
Rep: Thank you for your seriousness and carefulness. Zinc promotes neurogenesis [13], improves immunity [14], acts as a cofactor for enzymes [15], and participates in synaptic transmission [16] and cell apoptosis [17]. We have deleted that zinc plays two roles, because it doesn't convey the right meaning. What we want to state is the following. When synaptic zinc is in a homeostatic state, preventive zinc supplementation can improve the body's immunity and reduce the production of inflammatory factors in the brain and is conducive to the utilization of zinc in the body [18, 19]. Under intense stress conditions such as cerebral ischemia [3], epilepsy [20], and hypoglycemia [21], the presynaptic membrane Zn2+ is released at a great rate into the synaptic cleft, resulting in Zn2+ overload in the postsynaptic neurons and neurotoxicity.
- Higashi, Y., T. Aratake, S. Shimizu, T. Shimizu, K. Nakamura, M. Tsuda, T. Yawata, T. Ueba, and M. Saito. "Influence of Extracellular Zinc on M1 Microglial Activation." Sci Rep 7 (2017): 43778.
- Mortadza, S. S., J. A. Sim, M. Stacey, and L. H. Jiang. "Signalling Mechanisms Mediating Zn(2+)-Induced Trpm2 Channel Activation and Cell Death in Microglial Cells." Sci Rep 7 (2017): 45032.
- Ueba, Y., T. Aratake, K. I. Onodera, Y. Higashi, T. Hamada, T. Shimizu, S. Shimizu, T. Yawata, R. Nakamura, T. Akizawa, T. Ueba, and M. Saito. "Attenuation of Zinc-Enhanced Inflammatory M1 Phenotype of Microglia by Peridinin Protects against Short-Term Spatial-Memory Impairment Following Cerebral Ischemia in Mice." Biochem Biophys Res Commun 507, no. 1-4 (2018): 476-83.
- Lai, C., Z. Chen, Y. Ding, Q. Chen, S. Su, H. Liu, R. Ni, and Z. Tang. "Rapamycin Attenuated Zinc-Induced Tau Phosphorylation and Oxidative Stress in Rats: Involvement of Dual Mtor/P70s6k and Nrf2/Ho-1 Pathways." Front Immunol 13 (2022): 782434.
- Moriyama, M., S. Fujitsuka, K. Kawabe, K. Takano, and Y. Nakamura. "Zinc Potentiates Lipopolysaccharide-Induced Nitric Oxide Production in Cultured Primary Rat Astrocytes." Neurochem Res 43, no. 2 (2018): 363-74.
- Zhang, L., Q. Guo, Y. Duan, X. Lin, H. Ni, C. Zhou, and F. Li. "Comparison of the Effects of Inorganic or Amino Acid-Chelated Zinc on Mouse Myoblast Growth in Vitro and Growth Performance and Carcass Traits in Growing-Finishing Pigs." Front Nutr 9 (2022): 857393.
- Fazel Torshizi, F., M. Chamani, H. R. Khodaei, A. A. Sadeghi, S. H. Hejazi, and R. Majidzadeh Heravi. "Therapeutic Effects of Organic Zinc on Reproductive Hormones, Insulin Resistance and Mtor Expression, as a Novel Component, in a Rat Model of Polycystic Ovary Syndrome." Iran J Basic Med Sci 23, no. 1 (2020): 36-45.
- Maares, M., C. Keil, L. Pallasdies, M. Schmacht, M. Senz, J. Nissen, H. Kieserling, S. Drusch, and H. Haase. "Zinc Availability from Zinc-Enriched Yeast Studied with an in Vitro Digestion/Caco-2 Cell Culture Model." J Trace Elem Med Biol 71 (2022): 126934.
- Zhang, Z., Q. Cheng, Y. Liu, C. Peng, Z. Wang, H. Ma, D. Liu, L. Wang, and C. Wang. "Zinc-Enriched Yeast May Improve Spermatogenesis by Regulating Steroid Production and Antioxidant Levels in Mice." Biol Trace Elem Res 200, no. 8 (2022): 3712-22.
- da Costa, B., B. Pippi, S. J. Berlitz, A. R. Carvalho, M. L. Teixeira, I. C. Külkamp-Guerreiro, S. F. Andrade, and A. M. Fuentefria. "Evaluation of Activity and Toxicity of Combining Clioquinol with Ciclopirox and Terbinafine in Alternative Models of Dermatophytosis." Mycoses 64, no. 7 (2021): 727-33.
- da Costa, B., B. Pippi, T. F. Andrzejewski Kaminski, S. F. Andrade, and A. M. Fuentefria. "In Vitro Antidermatophytic Synergism of Double and Triple Combination of Clioquinol with Ciclopirox and Terbinafine." Mycoses 63, no. 9 (2020): 993-1001.
- Feng, L., J. Gao, Y. Wang, Y. K. Cheong, G. Ren, and Z. Yang. "Etidronate-Zinc Complex Ameliorated Cognitive and Synaptic Plasticity Impairments in 2-Vessel Occlusion Model Rats by Reducing Neuroinflammation." Neuroscience 390 (2018): 206-17.
- Choi, B. Y., I. Y. Kim, J. H. Kim, A. R. Kho, S. H. Lee, B. E. Lee, M. Sohn, J. Y. Koh, and S. W. Suh. "Zinc Transporter 3 (Znt3) Gene Deletion Reduces Spinal Cord White Matter Damage and Motor Deficits in a Murine Mog-Induced Multiple Sclerosis Model." Neurobiol Dis 94 (2016): 205-12.
- Feijó, G. D. S., J. Jantsch, L. L. Correia, S. Eller, O. V. Furtado-Filho, M. Giovenardi, M. Porawski, E. Braganhol, and R. P. Guedes. "Neuroinflammatory Responses Following Zinc or Branched-Chain Amino Acids Supplementation in Obese Rats." Metab Brain Dis 37, no. 6 (2022): 1875-86.
- Lin, P. H., M. Sermersheim, H. Li, P. H. U. Lee, S. M. Steinberg, and J. Ma. "Zinc in Wound Healing Modulation." Nutrients 10, no. 1 (2017).
- Sensi, S. L., P. Paoletti, A. I. Bush, and I. Sekler. "Zinc in the Physiology and Pathology of the Cns." Nat Rev Neurosci 10, no. 11 (2009): 780-91.
- Li, X., S. Chen, L. Mao, D. Li, C. Xu, H. Tian, and X. Mei. "Zinc Improves Functional Recovery by Regulating the Secretion of Granulocyte Colony Stimulating Factor from Microglia/Macrophages after Spinal Cord Injury." Front Mol Neurosci 12 (2019): 18.
- de Oliveira, S., G. D. S. Feijó, J. Neto, J. Jantsch, M. F. Braga, Lfds Castro, M. Giovenardi, M. Porawski, and R. P. Guedes. "Zinc Supplementation Decreases Obesity-Related Neuroinflammation and Improves Metabolic Function and Memory in Rats." Obesity (Silver Spring) 29, no. 1 (2021): 116-24.
- Mousaviyan, R., N. Davoodian, F. Alizadeh, M. Ghasemi-Kasman, S. A. Mousavi, F. Shaerzadeh, and H. Kazemi. "Zinc Supplementation During Pregnancy Alleviates Lipopolysaccharide-Induced Glial Activation and Inflammatory Markers Expression in a Rat Model of Maternal Immune Activation." Biol Trace Elem Res 199, no. 11 (2021): 4193-204.
- Lee, J. Y., J. H. Kim, R. D. Palmiter, and J. Y. Koh. "Zinc Released from Metallothionein-Iii May Contribute to Hippocampal Ca1 and Thalamic Neuronal Death Following Acute Brain Injury." Exp Neurol 184, no. 1 (2003): 337-47.
- Kho, A. R., B. Y. Choi, J. H. Kim, S. H. Lee, D. K. Hong, S. H. Lee, J. H. Jeong, M. Sohn, and S. W. Suh. "Prevention of Hypoglycemia-Induced Hippocampal Neuronal Death by N-Acetyl-L-Cysteine (Nac)." Amino Acids 49, no. 2 (2017): 367-78.
Reviewer 2 Report
The article: ”Zinc homeostasis: an emerging therapeutic target for neuroinflammation related diseases” covers all the main facets of zinc homeostasis in the central nervous system and the development of neuroinflammation. Still, there are some aspects that require more profound attention.
Revision suggestions look for:
Minor:
1. Lack of citations in lines 46-50
2. Lack of citations in lines 51-64
Major:
1. Support of clinical studies - references
2. Limitations in restoring zinc homeostasis possibilities
3. Possibilities of zinc level monitoring
Author Response
Minor:
- Lack of citations in lines 46-50
- Lack of citations in lines 51-64
Rep: We have added the references.
Major:
- Support of clinical studies – references
Rep: Thank you for your advice. This is an important issue that deserves continued attention and research. The number of relevant clinical literatures is small. Clinical studies have shown that the onset of depression and epilepsy is related to the concentration of Zn2+. Clinical research has shown that the decrease in zinc concentration in peripheral blood is related to the occurrence and development of depression [1], and zinc supplementation enhances the therapeutic effect of antidepressants [2-4]. There were significantly reduced serum concentrations of zinc and selenium in patients with febrile seizures compared with control [5]. These references have been cited in our article.
- Swardfager, W., N. Herrmann, G. Mazereeuw, K. Goldberger, T. Harimoto, and K. L. Lanctôt. "Zinc in Depression: A Meta-Analysis." Biol Psychiatry 74, no. 12 (2013): 872-8.
- da Silva, L. E. M., M. L. P. de Santana, P. R. F. Costa, E. M. Pereira, C. M. M. Nepomuceno, V. A. O. Queiroz, L. P. M. de Oliveira, Mepdc Machado, and E. P. de Sena. "Zinc Supplementation Combined with Antidepressant Drugs for Treatment of Patients with Depression: A Systematic Review and Meta-Analysis." Nutr Rev 79, no. 1 (2021): 1-12.
- Siwek, M., D. Dudek, I. A. Paul, M. Sowa-Kućma, A. Zieba, P. Popik, A. Pilc, and G. Nowak. "Zinc Supplementation Augments Efficacy of Imipramine in Treatment Resistant Patients: A Double Blind, Placebo-Controlled Study." J Affect Disord 118, no. 1-3 (2009): 187-95.
- Nowak, G., M. Siwek, D. Dudek, A. Zieba, and A. Pilc. "Effect of Zinc Supplementation on Antidepressant Therapy in Unipolar Depression: A Preliminary Placebo-Controlled Study." Pol J Pharmacol 55, no. 6 (2003): 1143-7.
- Saghazadeh, A., M. Mahmoudi, A. Meysamie, M. Gharedaghi, G. W. Zamponi, and N. Rezaei. "Possible Role of Trace Elements in Epilepsy and Febrile Seizures: A Meta-Analysis." Nutr Rev 73, no. 11 (2015): 760-79.
- Limitations in restoring zinc homeostasis possibilities
Rep: This is a good question. Firstly, many preparations for zinc supplementation have been used in clinical practice, but few zinc chelating agent. Secondly, there are no very accurate clinical means and criteria to determine the brain zinc concentration. Thirdly, numerous clinical trials will be needed to verify whether restoration of zinc homeostasis is neuroprotective.
- Possibilities of zinc level monitoring
Zinc level monitoring: Serum, hair, and urinary metal and metalloid analysis were performed by inductively coupled plasma mass spectrometry at NexION 300D (PerkinElmer Inc., USA)[1].
- Detection of zinc in serum:
1.1 Atomic absorption spectroscopy method had been used to determine zinc in patient serum samples [2, 3].
1.2 Ion chromatography was used to detect levels of zinc in blood plasma [4, 5].
1.3 Plasma levels of zinc was measured by atomic absorption spectrophotometer and electrochemiluminescence methods, respectively[6].
- Determination of zinc in cerebrospinal fluid:
The content of zinc in cerebrospinal fluid (CSF) of normal persons and the patients were determined directly by atomic absorption spectrometry[7].
- Determination of zinc in nerve cell:
Changes in intracellular Zn2+ levels were tracked in individual neurons by microflfluorometry using the Zn2+ selective flfluorophore, FluoZin3[8].
- In addition to some conventional detection methods, the concentration of zinc can also be detected by designing fluorescent probes to observe the fluorescence intensity.
4.1 A deep-red-fluorescent zinc probe (JJ) was designed and synthesized with a membrane-targeting cholesterol unit. The addition of ZnCl2 elicits an approximately 5-fold enhancement of the fluorescence emission with a fluorescence dynamic range of 141000.JJ was found to be nontoxic which enables cellular imaging[9].
4.2 Visual detection of Zn2+ makes use of the chelation of Zn2+ and probes to improve the fluorescence quantum yield of fluorescent groups. A turn-on flfluorescence probe for Zn2+ had been synthesized, which contained a symmetric pseudo salen moiety as zinc recognition site.A linear relationship was observed, which enabled the quantitative determination of Zn2+ [10].
4.3 By attaching fluorescent zinc probes to the plasma membrane of pancreatic cells, insulin secretion can also be tracked.Developing a fluorescent, cell surface-targeted zinc indicator for monitoring induced exocytotic release (ZIMIR). ZIMIR displayed a robust fluorescence enhancement on Zn2+ chelation and bound Zn2+ with high selectivity.When added to the culture β Cells or intact islets,ZIMIR labeled cells measured the change of Zn2+ concentration near the fusion site[11].
We do not cite them in our article, but the problem makes us think hard.
- Tinkov, A. A., M. G. Skalnaya, O. P. Ajsuvakova, E. P. Serebryansky, J. C. Chao, M. Aschner, and A. V. Skalny. "Selenium, Zinc, Chromium, and Vanadium Levels in Serum, Hair, and Urine Samples of Obese Adults Assessed by Inductively Coupled Plasma Mass Spectrometry." Biol Trace Elem Res 199, no. 2 (2021): 490-99.
- Krishna, P. G., K. S. Rao, O. B. Devi, and G. R. Naidu. "Analysis of Samples of Human Serum with Cataracts for Zinc and Iron by Flame Atomic Absorption Spectrometry." Indian J Environ Health 45, no. 3 (2003): 189-94.
- Hu, J., Y. M. Chang, S. B. Gao, C. X. Hai, J. S. Li, and X. P. Xie. "[Speciation Analysis of Trace Elements Cu, Fe and Zn in Serum by Flame Atomic Absorption Spectrophotometry]." Guang Pu Xue Yu Guang Pu Fen Xi 28, no. 3 (2008): 700-3.
- Lane, E., A. J. Holden, and R. A. Coward. "Determination of Copper and Zinc in Blood Plasma by Ion Chromatography Using a Cobalt Internal Standard." Analyst 124, no. 3 (1999): 245-9.
- Ong, C. N., H. Y. Ong, and L. H. Chua. "Determination of Copper and Zinc in Serum and Whole Blood by Ion Chromatography." Anal Biochem 173, no. 1 (1988): 64-9.
- Konukoglu, D., M. S. Turhan, M. Ercan, and O. Serin. "Relationship between Plasma Leptin and Zinc Levels and the Effect of Insulin and Oxidative Stress on Leptin Levels in Obese Diabetic Patients." J Nutr Biochem 15, no. 12 (2004): 757-60.
- Sun, R. X., Y. X. Su, and J. H. Sun. "[Determination of Trace Elements in Cerebrospinal Fluid (Csf) of Patients Suffering Cerebrovascular Disease by Atomic Absorption Spectrometry]." Guang Pu Xue Yu Guang Pu Fen Xi 26, no. 4 (2006): 720-2.
- Qin, Y., D. Thomas, C. P. Fontaine, and R. A. Colvin. "Silencing of Znt1 Reduces Zn2+ Efflux in Cultured Cortical Neurons." Neurosci Lett 450, no. 2 (2009): 206-10.
- Kim, J. J., J. Hong, S. Yu, and Y. You. "Deep-Red-Fluorescent Zinc Probe with a Membrane-Targeting Cholesterol Unit." Inorg Chem 59, no. 16 (2020): 11562-76.
- Yu, H., T. Yu, M. Sun, J. Sun, S. Zhang, S. Wang, and H. Jiang. "A Symmetric Pseudo Salen Based Turn-on Fluorescent Probe for Sensitive Detection and Visual Analysis of Zinc Ion." Talanta 125 (2014): 301-5.
- Li, D., S. Chen, E. A. Bellomo, A. I. Tarasov, C. Kaut, G. A. Rutter, and W. H. Li. "Imaging Dynamic Insulin Release Using a Fluorescent Zinc Indicator for Monitoring Induced Exocytotic Release (Zimir)." Proc Natl Acad Sci U S A 108, no. 52 (2011): 21063-8.
Round 2
Reviewer 1 Report
The author has precisely addressed my comments. I have no additional comments.
Reviewer 2 Report
All my comments have been taken into account. The article has been greatly improved, and I consider it can now be accepted in its current form.